# Validating the Well-Being of Older People (WOOP) Instrument in China

**DOI:** 10.3390/ijerph20010277

**Published:** 2022-12-24

**Authors:** Runhua Liu, Zhuxin Mao, Zhihao Yang

**Affiliations:** 1Department of Health Services Management, Guizhou Medical University, Guiyang 550025, China; 2Center of Medicine Economics and Management Research, Guizhou Medical University, Guiyang 550025, China; 3Centre for Health Economics Research and Modelling Infectious Diseases, Vaccine and Infectious Disease Institute, University of Antwerp, 2000 Antwerp, Belgium

**Keywords:** well-being, older people, utility, rural areas, China

## Abstract

Generic health-related quality of life (HRQoL) measures have been used for estimating utility value, which is then used for calculating quality-adjusted life years (QALYs). HRQoL measures may not capture many of the relevant and important non-health aspects of quality of life. The well-being of older people (WOOP) instrument was first developed in the Netherlands. This study aimed to validate this new instrument among older people in China. WOOP was first translated into simplified Chinese (for use in Mainland China) by two experienced translators. From July to August 2022, a cross-sectional study was conducted on a convenience sample of 500 older people in Southwestern China. Older people who provided consent reported their demographic information and completed the simplified Chinese version of the WOOP instrument using a pencil and paper. The feasibility of WOOP was determined by the percentage of missing responses. Then, using the data without any missing responses, we examined the item response distributions, pairwise Spearman correlations, underlying factors, and known-group validity of WOOP. Among the nine items of WOOP, three had more than 10% missing responses. The response distributions of the nine items were overall good without signs of ceiling and floor effects. The correlations among the WOOP items were low. A two-factor exploratory factor analysis model suggested that the WOOP items can be categorized into either internal or external well-being items. Good known-group validity results were found. Some WOOP items may not be easily understood by a small proportion of rural residents. However, other results have suggested WOOP to be a valid instrument for measuring the well-being of the elderly in China. The availability of WOOP enables the measurement of well-being-related utility.

## 1. Background

The number and proportion of older people worldwide are growing rapidly. The 2020 Chinese census data showed that the proportion of people aged over 65 had reached 13.5% [1], indicating that China will soon transition from an aging into an aged society [2]. The rapidly aging Chinese population leads to an escalating demand for health and social care services [3,4]. Due to the finite amount of health and social care resources, economic evaluations are urgently needed to prioritize the decisions in health and social care services, comparing the costs and outcomes of alternate options.

Quality-adjusted life years (QALY), which combines both the quantity and quality of life, is the most frequently used outcome measure in health and social care settings [5], despite the criticism of its use [6]. Quality weights of a determined health state are needed to operationalize the QALY concept [5]. To obtain quality weights, generic health-related quality of life (HRQoL) measures, such as EuroQol five-dimension (EQ-5D), are widely used [5]. Because HRQoL focuses on the subjective perceptions of individuals’ own health status, it is not an objective measure. To appropriately reflect the actual health status of individuals, the quality of subjective measures is often validated by assessing their psychometric properties. The primary evaluation criteria for psychometric properties include validity and reliability. Most of the HRQoL measures were developed based on the conceptual framework of health [7]. The aim of the health and social care services for the elderly is not only to improve the health of older people but more broadly to improve their well-being [8,9]. HRQoL measures may not capture many of the relevant and important non-health aspects of quality of life.

Considering the limitation with most generic HRQoL measures, various well-being measures have been developed and have the potential to estimate QALY for use across long-term, social, and palliative care settings. The recently developed well-being of older people (WOOP) measure has distinct advantages among the available well-being measures [10]. First, the development of this measure fully considered the important concepts of the well-being of the elderly [10]. The conceptual framework of this measure was constructed based on an in-depth investigation of the understanding of the well-being of a diverse sample of older populations [11]. Furthermore, this measure is neither too lengthily nor too narrowly defined; therefore, it is well-suited for self-completion. WOOP incorporates a comprehensive list of well-being domains including physical health, mental health, social life, receiving support, acceptance and resilience, feeling useful, independence, making ends meet, and living situation [10]. Both qualitative and quantitative validation studies have been conducted to prove that the measure captures all relevant and important domains to quantify well-being in the target populations [10,12]. Additionally, utility tariffs have been estimated in the Netherlands [13], enhancing the potential of this measure for economic evaluations in health and social care services for the elderly.

With the growing need to rationalize resource allocation in health and social care in China, it is necessary to obtain a well-designed well-being measure for use among the older Chinese population. However, in China, limited well-being measures are available. The Chinese version of the ICEPOP Capability measure for adults (ICECAP-A) was adapted and validated [14] but has not been frequently used in practice since its adaption. Empirical evidence has also shown that ICECAP-A performed less satisfactorily compared with HRQoL measures such as EQ-5D and Assessment of Quality of Life (AQoL)-7D in a Chinese population [15,16]. The cross-culturally developed well-being measure, EQ Health and Well-being (EQ-HWB), included Chinese participants in its development process, but its conceptual framework needs further validation in China [17]. The newly developed WOOP can be an alternative to assessing well-being among the Chinese elderly. Until now, the measurement properties of WOOP have only been reported in a Dutch population. Due to cultural differences in perceiving health and well-being, it is essential to test the appropriateness of this measure in Chinese populations, whose cultures are significantly different from the Western culture [18].

Although the rural population occupies a large proportion of the total populations in low- and middle-income countries (67% and 46%, respectively), there has been disproportionately less research on the comprehension and feasibility of health outcome measures among such populations in underdeveloped regions. Specifically, in China, although the rural population comprises 37% of the population (about 0.56 billion people) [19], the Chinese versions of EQ-5D were mostly validated among the general Chinese population in urban communities with generally high educational attainment [20]. Therefore, little is known about the suitability of the instruments among the people living in rural areas in China. Additionally, among the three Chinese national valuation studies, two studies (one calculating the EQ-5D-3L value set and the other calculating the EQ-5D-5L value set) included only urban Chinese residents [21], and the understanding and preferences of rural people regarding health are more likely to be neglected. On the contrary, significant disparities have been found in health and well-being between the aging population living in rural and urban China due to the inequality in the access to and the quality of healthcare services [22]. Measuring and tracking the well-being status of the older population in rural areas is essential for informing policy decisions and resource allocation, which will promote well-being and address health inequality. To achieve this, a validated measure among the older population in rural areas is needed.

This study, therefore, aimed to introduce WOOP for use in measuring the well-being of the older Chinese population in Chinese rural areas and to translate and validate the Chinese version of this measure.

## 2. Method

### 2.1. Sampling and Questionnaire

The data was collected in the rural areas of Southwestern China, including Yunnan Province and Guizhou Province. We targeted older people (>60 years old) who live in rural areas in these two provinces. Other inclusion criteria included respondents who were registered as rural residents and had lived in a rural area over the past year, were either able to read the questionnaire or able to converse in the local language, and who gave informed consent. This study was approved by the Ethics Committee of Guizhou Medical University (Approval letter No. 276-2022).

The questionnaire consisted of a set of demographic questions including sex (male and female), age groups (60–64, 65–74, 75–79, or ≥80 years), marital status, living status (living with a partner, living with offspring, or living alone), area of residence (village or township), health condition (healthy or with at least one health condition). In China’s administrative structure, a township is higher than a village and is often more urbanized than a village. The classification of age groups followed the consensus of statistics on China’s aging population: young-elderly group (aged 60–64 years), middle-elderly group (aged 65–79 years), and highly aged group (aged 80 and above years) [23]. Nine WOOP items were included in the questionnaire after the questions on demographics. At the end of the questionnaire, the participants reported their health conditions (healthy or with at least one health condition). In total, we set the sample size at 500 and printed out 500 questionnaires for this study.

### 2.2. Simplified Chinese Version of WOOP

Two experienced translators whose native language is Mandarin and are fluent in English independently translated the WOOP instrument into simplified Chinese (for use in Mainland China). Two of the study authors discussed the translations and decided on the final wording of the WOOP instrument.

In general, WOOP responses can be organized and analyzed like existing multi-attribute utility instruments (MAUIs), such as EQ-5D [24,25], which is the most widely used MAUI. WOOP has nine items, each with five response levels corresponding to excellent, good, fair, poor, and bad performances and coded as 1, 2, 3, 4, and 5, respectively. The level summary score (LSS) representing the WOOP score is between 9 (corresponding to the state 111111111) and 45 (corresponding to the state 555555555). The WOOP state can also be converted into utility on a QALY scale, and currently, only one value set has been established [13].

### 2.3. Data Collection Procedure

The convenience sampling method was used and the data was collected from July to August 2022. Five undergraduate students were recruited from Guizhou Medical University as interviewers. All the interviewers received standardized data collection training on the background of this study, and were introduced to the questionnaire, including the collection of demographic information and the WOOP item responses. The questionnaire was filled out using a pencil and paper.

In each sampling location, we recruited participants through the local government. Upon agreeing to help, a local government official introduced the interviewers to the participants at their homes and then explained the purpose of this study. An oral informed consent was obtained before the respondents completed the questionnaire. The respondents were encouraged to independently respond to the questionnaire, and the interviewers were allowed to explain the questions, if necessary. For those with vision problems or who were illiterate, the investigators switched to an interview mode by reading out the questions for the respondents. Respondents could choose to skip a question if they had difficulties understanding it or did not feel like answering it. Participants could discontinue their participation at any time during data collection. Upon completion of the questionnaire, each respondent received a gift (e.g., a towel) as an incentive. All questionnaires were sent to Guizhou Medical University in September, and the data were inputted on a Microsoft Excel (Microsoft Cooperation 2018, Redmond, Washington) spreadsheet by the interviewers.

### 2.4. Data Analysis

We analyzed and reported: 1) feasibility, which was measured by the number of missing responses for each item, and we examined whether feasibility differed by demographics; 2) item response distributions, including whether each item had ceiling and floor effects; 3) pairwise Spearman correlations among all items; 4) exploratory factor analysis (EFA) results; and 5) known-group validity. For 2) and 5), the analysis was performed on both item level (i.e., item response) and aggregate level (i.e., LSS and utility). We calculated utility based on the value set of the Netherlands [13].

First, the frequency percentage of missing responses was reported for each item. It is worth noting that from this step onward, we only used the data without any missing responses since both LSS and utility cannot be calculated with any missing data. Next, we examined the data distribution of all items to identify whether all five response levels were used. We checked whether ceiling [26] and floor effects were present, i.e., items with over 70% of responses using level 1 (excellent) and level 5 (bad), suggesting ceiling effect and floor effect, respectively [17]. On the aggregate level, we also plotted histograms for both LSS and utility. Next, we examined the correlation between all items by computing pairwise Spearman correlation coefficients using the following criteria to define correlation strength: trivial: <0.10, small: 0.10–0.29, moderate: 0.30–0.49, high: 0.50–0.69, very high: 0.70–0.89, and perfect: >0.90 [10]. High and perfect correlations suggested that the items may be measuring the same construct [17]. EFA was conducted to further understand the relationship among the items. Prior to the EFA analysis, we conducted a Bartlett’s test for sphericity and calculated the Kaiser–Meyer–Olkin (KMO) measure of sampling adequacy. A parallel analysis was performed to determine the number of factors. Only the highest factor loading was reported for each item, which needed to be ≥0.40 [10]. We reported the LSS and utility for our entire sample and by demographic subgroups, for example, by sex (male vs. female). Lastly, the Mann–Whitney U test, Cohen’s D effect size, and chi-square tests were used to assess whether the LSS, utility, and each item could differentiate respondents with different demographic backgrounds. In this step, we assumed males, the elderly (age groups were merged into two groups to an achieve equivalent sample size for each group), widowed respondents who lived alone in rural areas, and respondents who reported at least one health condition would report more problems both overall and for each dimension and that these respondents would also have lower utility. For both the Mann–Whitney U and chi-square tests, a *p*-value was set at 0.05. The Cohen’s D effect size was compared following the defined criteria: 0.2 to 0.5 suggested small, 0.5 to 0.8 suggested medium, and ≥0.8 suggested large effect sizes. In addition, we examined whether the distribution of item scores was different between the sample without and with missing data (Appendix A), and we examined how demographic characteristics could influence the responses to the WOOP items (Appendix A).

## 3. Results

A total of 500 questionnaires were distributed and collected. Of these, 26 respondents dropped out during the interview as they had difficulties differentiating the response levels; 121 questionnaires were completed with some missing data; and 353 questionnaires were completed without any missing data. The questionnaires answered by females were more likely to have missing data (chi-square test, *p* < 0.05). Table 1 shows that demographic information of the sample according to missing or no missing data. With all the demographic information collected, the sample could further be divided into different sub-groups, e.g., 46.74% of the sample lived in counties, and 43.91% of the sample had at least one health condition. Note our sample was not representative of the general population in China.

“Physical health,” “making ends meet,” and “living situation” did not have any missing responses. The frequency and percentage of missing responses by item, from the highest to the lowest, were 64 (13.5%) for “receive support,” 61 (12.9%) for “independence,” 51 (10.8%) for “acceptance and resilience,” 19 (4.0%) for “social contacts,” 8 (1.7%) for “feeling useful,” and 2 (0.4%) for “mental health.” Females were more likely to skip responses to the questions and have more missing responses.

### 3.1. Data Distribution

Figure 1 shows the response distributions of all the nine items of the WOOP measure. Overall, level 2 (good) and level 3 (fair) responses accounted for the largest proportion of responses, and all five levels of responses were used effectively, with differences among the items. We did not observe any ceiling and floor effects for all the items based on the predefined criteria. “Mental health” had the highest proportion (173, 49.1%) of an excellent response, followed by “physical health” (111, 31.4%) and “receive support” (61, 17.3%). “Making ends meet” had the highest proportion (58, 16.4%) of a bad response, followed by “independence” (43, 12.2%) and “acceptance and resilience” (22, 6.2%).

Table 2 shows the Spearman correlations among the WOOP items. No “very high” and “perfect correlations” were observed, and the correlations were between “small” and “high”, except for a trivial correlation observed between “receive support” and “independence.” “Physical health” showed the most correlation with other items including “mental health”, “acceptance and resilience”, “independence”, and “making ends meet.” “Acceptance”, “feeling useful”, and “independence” showed high correlations with each other. Both “receive support” and “living situation” did not show any high correlations with the other items.

### 3.2. Exploratory Factor Analysis

Bartlett’s test of sphericity showed a *p*-value of 0.00 and a KMO value of 0.872, which implied that the data is suitable for factor analysis. The parallel analysis suggested a two-factor model, and Table 3 shows the EFA results. The first factor included “physical health”, “mental health”, “social contacts”, “acceptance and resilience”, ”feeling useful”, “independence”, and “making ends meet”; the second factor included “social contacts”, “receive support”, and “living situation.” The two factors (1 and 2) explained 76.8% and 27.6% of variance, respectively.

### 3.3. Known-Group Validity

For the six known groups defined in this study, all the results were not significant for sex and residence area (Table 4). For the other four known groups, both LSS and utility could differentiate the respondents; particularly, for the respondents with health condition, the effect size was 1.36, implying a strong effect. The effect sizes for age and marital and living statuses were moderate. On the item level, respondents with a health condition compared with those without a health condition, reported more significant problems in all the nine items. “Receive support” and “living situation” were not significantly different by age, marital status, and living status. Moreover, “mental health” and “social contacts” were not significantly different by living status.

## 4. Discussion

In this study, we translated the WOOP instrument into simplified Chinese and tested its use among rural older individuals in Southwestern China. Until now, WOOP, a newly developed instrument, had only been validated in a Dutch population and had not yet been tested in a non-Western setting. Despite cultural differences in perceiving health and wellbeing [28,29], this study suggests WOOP to be a valid instrument for measuring the well-being of the elderly in China and supports its use in a non-Western setting. In addition, we focused on the rural population, whose voice has often been neglected when validating and valuing a HRQoL or well-being measure. This study enriches evidence on the comprehension and feasibility of well-being measures among rural populations in underdeveloped areas.

Approximately 30% of the respondents in our study had difficulty understanding at least one item in the WOOP instrument and could not provide a response. The top three items with which the respondents had difficulties were “receive support”, “independence”, and “acceptance.” This suggests that older Chinese people living in rural areas may not have a clear understanding of self-worth and social relationship, probably because these items, compared with those without missing responses, i.e., “physical health”, “living situation”, and “making ends meet” tend to measure more vaguely defined constructs. Similarly, previous studies have reported that older Chinese rural people had more difficulty in comprehending anxiety/depression than physical functioning items (mobility, self-care, and usual activities) in the EQ-5D [30,31]. This also indicates the need to better adapt or improve item descriptions, especially for the conceptually more abstract items in HRQoL and well-being measures for use among rural residents. Notably, in our study, interviewers were allowed to explain the meaning of each item to the respondents; therefore, more missing responses would be expected if WOOP is entirely self-administered by older people.

Other than the feasibility issue, the WOOP instrument showed good distributions and known-group validity results, suggesting that a large proportion of individuals could understand these items and effectively use the response levels to respond to the questionnaire. The response distributions of the WOOP items did not show signs of ceiling and floor effects, which have been reported by other HRQoL measures such as EQ-5D-5L and short-form six-dimension (SF-6D) [26,32]. A possible reason is that the WOOP instrument used “excellent” as the best response level and “good” as the second-best response level for most items, but EQ-5D-5L used “no problems” as the best response level. The percentage of respondents who selected the best level was highest for “physical health” and “mental health,” which both used “no problem” as the best response level, similar to that used for EQ-5D-5L. This design prevents the ceiling effect problems, but may not be ideal for calculating utility value, as observed with the Dutch value set, for which seven second level coefficients were not more than 0.02 on the utility scale [13]. This suggests that individuals found the first two levels similarly acceptable and did not attach utility decrements for being on the second level for these dimensions.

For the known-group validity, although studies reported that males and those who lived in a township reported better health statuses [33,34], it is unclear whether their well-being was better. In this study, we also did not find any difference between these two groups. For other known groups, WOOP demonstrated a clear validity in differentiating respondents; both the WOOP LSS score and utility showed good known-group validity. This is consistent with the WOOP validation study findings in the Netherlands, which demonstrated the discriminatory power of the WOOP instrument [10]. It also suggests the potential of the WOOP instruments for use as a utility measure in economic evaluations. Considering that utility values measured by different HRQoL instruments are distinct [32,35], we may expect a large utility difference, as WOOP measures well-being and EQ-5D-5L measures HRQoL. Future studies should conduct a head-to-head comparison between WOOP and other HRQoL instruments, e.g., EQ-5D-5L.

The two-factor model showed that the WOOP instrument tapped into two major factors. The first factor may be interpreted as internal well-being, which covers one’s health, autonomy, and self-value, but not relationship with others as “independence” had the highest loading and “social contacts” had the lowest loading. The second factor may be interpreted as external well-being as both “social contacts” and “received support” were about social interactions, and the living situation may be considered as one’s social status and whether those people like or dislike their neighbors. The dimensionality of the WOOP instrument in our study was different from that reported in Hackert et al.’s study [10]. In their study, they combined the items in EQ-5D and WOOP to develop a 3-factor model: physical functioning, mental functioning, and a factor containing WOOP-only items (“social contacts”, “receive support”, “acceptance and resilience”, “feeling useful”, and “living situation”) and demonstrated that WOOP can capture outcomes beyond health status. Since we did not include other preference-based HRQoL measures to explore the constructs underlying the WOOP items, future studies are needed to explore whether and to what extent WOOP can capture outcomes beyond health status in China. Such information is important to guide future studies on well-being measures, such as the WOOP, alone or alongside HRQoL measures, to quantify outcomes that can be used in economic evaluations.

With the aging Chinese population and the growing need to optimize resource allocation strategies in health and social care, a well-designed well-being measure is essential for use among the older Chinese population. Since well-being measures are limited in China, WOOP can be considered as an option to assess the well-being of the elderly Chinese. This instrument has the potential to quantify the well-being status of the older Chinese population when evaluating the intervention strategies being implemented among this population, which will optimize policymaking. As the instrument has now been validated in a rural population, it can also be useful for data collection to provide health policy guidance and reduce inequality, which will improve health in rural areas.

This study has some limitations that need to be addressed. First, we did not collect data on other MAUI instruments, such as EQ-5D. Second, we only collected data at one time point and could not assess the test–retest reliability of WOOP using the collected cross-sectional data. Even though participants expressed difficulty in understanding some of the WOOP items, we did not systematically record their difficulties. Future studies should use qualitative methods to report how participants understand such items. In addition, this survey did not include a comprehensive set of demographic information, such as health insurance and income and education levels, which are associated with self-reported health status. Since we targeted the public in this study, we did not collect detailed information regarding their health conditions. Future studies should investigate how these factors affect the WOOP responses and to validate WOOP in urban populations. Lastly, all participants were recruited using the convenience sampling method, which cannot achieve sample representativeness. The results of this study cannot be generalized to the Chinese urban population.

## 5. Conclusions

Overall: this study suggested that the WOOP instrument is valid for measuring the well-being of the elderly in rural China. Future studies should further validate this instrument’s psychometric properties, such as content validity and test–retest reliability. Studies are also needed to understand the relationships between WOOP and other HRQoL measures, e.g., EQ-5D-5L. As WOOP was developed for use during the economic evaluations of health and social care services, future studies could specifically test its use in these settings.

## Figures and Tables

**Figure 1 ijerph-20-00277-f001:**
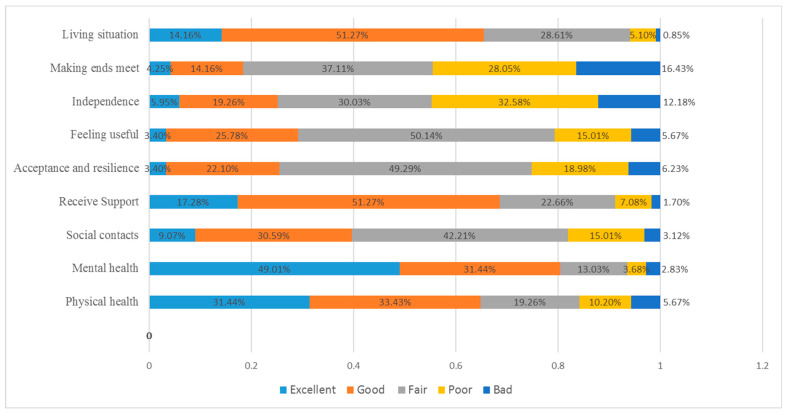
Response distributions (as percentage) by WOOP items, n = 353.

**Table 1 ijerph-20-00277-t001:** Demographic information of the sample, n = 474.

		Population Reference #	No Missing Data, n = 353		with Missing Data, n = 121	
		%	n	%	n	%
Sex	Female	48.25	170	48.16	75	61.98
	Male	51.75	183	51.84	46	38.02
Age group	61–64	27.79	87	24.65	36	29.75
	65–74	46.81	151	42.78	47	38.84
	75–79	11.83	87	24.65	23	19.01
	≥80	13.56	28	7.93	15	12.4
Marital status	Married	/	206	58.36	65	53.72
	Single	/	13	3.68	3	2.48
	Widowed	/	134	37.96	53	43.8
Living status	Living with partner	/	115	32.58	36	29.75
	Living with offspring	/	173	49.01	56	46.28
	living alone	/	65	18.41	29	23.97
Residence area	Township	/	165	46.74	46	38.02
	Village	/	188	53.26	75	61.98
Health status *	No health condition	/	198	56.09	/	/
	With at least one health condition	/	155	43.91	/	/

* Health status of those who had missing data were not recorded. # 2020 Population Census data [27].

**Table 2 ijerph-20-00277-t002:** Spearman correlations among WOOP items, n = 353.

	Physical Health	Mental Health	Social Contacts	Receive Support	Acceptance and Resilience	Feeling Useful	Independence	Making Ends Meet	Living Situation
Physical health									
Mental health	0.598								
Social contacts	0.469	0.521							
Receive Support	0.162	0.101	0.366						
Acceptance and resilience	0.510	0.460	0.443	0.137					
Feeling useful	0.469	0.423	0.476	0.247	0.698				
Independence	0.512	0.445	0.392	0.043	0.681	0.662			
Makes ends meet	0.532	0.452	0.403	0.081	0.601	0.558	0.672		
Living situation	0.246	0.255	0.338	0.295	0.137	0.230	0.260	0.312	

**Table 3 ijerph-20-00277-t003:** EFA results.

	**Item**	**Factor 1**	**Factor 2**	**Uniqueness**
	Physical health	0.631		0.495
	Mental health	0.597		0.511
	Social contacts	0.474	0.616	0.396
	Receive Support		0.512	0.737
	Acceptance and resilience	0.809		0.317
	Feeling useful	0.724		0.380
	Independence	0.823		0.310
	Making ends meet	0.746		0.402
	Living situation		0.417	0.779
Total variance explained		0.768	0.276	
Correlation with Factor 2		0.442		

**Table 4 ijerph-20-00277-t004:** Known-group validity results.

		n	LSS *	Utility	Physical Health	Mental Health	Social Contacts	Receive Support	Acceptance and Resilience	Feeling Useful	Independence	Making Ends Meet	Living Situation
			Mean, SD	Mean, SD	Mean, SD	Mean, SD	Mean, SD	Mean, SD	Mean, SD	Mean, SD	Mean, SD	Mean, SD	Mean, SD
Whole sample		353	23.89, 6.11	0.642, 0.317	2.25, 1.17	1.80, 0.99	2.73, 0.93	2.25, 0.88	3.03, 0.89	2.94, 0.88	3.26, 1.09	3.38, 1.05	2.27, 0.80
Sex	Female	170	24.39, 6.16	0.603, 0.333	2.37, 1.21	1.85, 1.04	2.72, 0.95	2.31, 0.88	3.03, 0.87	2.97, 0.88	3.35, 1.04	3.46, 1.00	2.32, 0.80
	Male	183	23.44, 6.06	0.644, 0.302	2.14, 1.12	1.75, 0.95	2.73, 0.92	2.19, 0.88	3.02, 0.91	2.90, 0.88	3.17, 1.12	3.31, 1.10	2.22, 0.79
				ES = −0.130									
Age group	60–74	238	22.55, 5.60	0.698, 0.261	2.05, 1.07	1.66, 0.86	2.58, 0.89	2.18, 0.89	2.82, 0.78	2.78, 0.82	3.05, 1.04	3.18, 0.98	2.24, 0.78
	≥75	115	26.69, 6.22	0.472, 0.368	2.67, 1.25	2.07, 1.17	3.02, 0.95	2.37, 0.85	3.45, 0.96	3.26, 0.90	3.70, 1.06	3.79, 1.09	2.34, 0.83
				ES = 0.754									
Marital status	Married	206	22.83, 5.13	0.692, 0.259	2.11, 1.08	1.67, 0.89	2.57, 0.86	2.27, 0.89	2.87, 0.72	2.83, 0.73	3.07, 0.97	3.23, 0.93	2.21, 0.77
	Single or widowed	147	25.39, 7.02	0.530, 0.366	2.46, 1.26	1.99, 1.10	2.94, 0.99	2.21, 0.88	3.24, 1.06	3.10, 1.03	3.52, 1.19	3.59, 1.17	2.35, 0.83
				ES = 0.526									
Living status	Living with others	288	24.60, 5.98	0.593, 0.330	2.38, 1.19	1.86, 1.03	2.77, 0.95	2.28, 0.87	3.14, 0.86	3.03, 0.86	3.34, 1.05	3.51, 1.02	2.28, 0.77
	Living alone	65	20.80, 5.78	0.762, 0.202	1.69, 0.86	1.52, 0.73	2.52, 0.85	2.08, 0.91	2.52, 0.85	2.54, 0.85	2.88, 1.17	2.83, 1.01	2.22, 0.93
				ES = −0.544									
Residence area	Township	165	23.40, 5.96	0.643, 0.293	2.30, 1.25	1.72, 0.95	2.59, 0.89	2.30, 0.93	2.92, 0.87	2.84, 0.84	3.19, 1.03	3.33, 1.13	2.21, 0.75
	Village	188	24.34, 6.23	0.608, 0.337	2.21, 1.10	1.87, 1.03	2.85, 0.95	2.20, 0.84	3.12, 0.91	3.02, 0.90	3.32, 1.13	3.43, 0.98	2.32, 0.84
				ES = 0.110									
Health status	No condition	198	20.74, 4.61	0.782, 0.170	1.51, 0.63	1.40, 0.67	2.39, 0.85	2.11, 0.81	2.69, 0.79	2.65, 0.75	2.88, 0.99	2.98, 0.95	2.12, 0.71
	With condition	155	27.94, 5.38	0.424, 0.348	3.20, 1.00	2.30, 1.11	3.15, 0.85	2.41, 0.95	3.46, 0.83	3.31, 0.89	3.74, 1.01	3.89, 0.96	2.46, 0.86
				ES = 1.360									

LSS: level summary score. * To test for differences, the Mann–Whitney U test was used for LSS, Cohen’s D effect size was used for utility, and the chi-square test was used for the items. For Cohen’s D effect size, effect sizes are reported; for the Mann–Whitney U and chi-square tests, shaded coefficients suggest statistical significance at 0.05.

## Data Availability

Available upon request to the corresponding author.

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
