# Peer review of "Validating the Well-Being of Older People (WOOP) Instrument in China"

_ijerph, 2022, doi:10.3390/ijerph20010277_

Round 1

Reviewer 1 Report

Thank you for the opportunity to review this paper.

While the article is short, the authors would do well to set up a clear agenda for the reader. They should state where they are taking the reader and why it is novel/important.

I would like the authors to give more information on the appropriateness or trustworthiness of QALYs, specifically why it is okay to combine quantity and quality of life as a measure(s). 

More justification could be offered regarding why the authors chose rural contexts for this study of older adults. 

I would appreciate if the fonts of the graphics match the style of the article. Figure 1 is labeled spearman correlations. That should be fixed.

The conclusion and discussion needs much more in the novelty of the article. What should readers want to do next regarding research, policy, and practice. Why is the Chinese context unique? Why is the rural context unique? 

Reviewer 3 Report

The paper was interesting and very clearly written. I was particularly fascinated by the spearman correlations.

My main comments

a) the age groups. On l. 97 4 of different sizes are mentioned, on l. 101 different bands and table 4. I suggest 5 year bands, though l. 101 is interesting to include. A comment is least needed with the known-group validity to explain why you split it here.

b) the known-validity is not entirely clear. I would have appreciated a short explanation of exactly which values are in each column. Is the 0 actually < 0.01? In the text. a brief explanation about the measures, e.g. which limits are used here if larger is better/worse, etc. 

With the discussion/interpretation/next steps, points that occurred to me:

* The physical health of women was lower. How did their age compare?

* With married couples, did you interview only one? i.e. could this have influenced the results.

* Towards the beginning (l 79), it mentions cultural aspects.

Re Abstract:

* I suggest adding the information that it was new/translated as part of this research.

* l. 18: I found the expression "simplified" confusing here - also because I didn't notice it elsewhere. 

A few small language mistakes I noticed:

l. 17: "Consented older people" -> Older people who gave their consent?

l. 55: "fully" -> full

l. 199: "prefect" -> perfect

Table 3: "Makeing" -> Making

l. ?? Discussion last line of 2nd paragraph: "disutility". Word not common and not sure what was meant.
